# When States Regionally Integrate: How Relative Economic Size and Institutional Homogeneity Matter

Gaspare M. Genna 

Department of Political Science and Public Administration, The University of Texas at El Paso, El Paso, TX 79968, USA; ggenna@utep.edu

**Abstract:** This study compares two articles that seek to explain why states participate in regional integration organizations and why they want to deepen their economic and political partnerships. The method of comparison is the systemist diagrammatic approach, which requires a deconstruction and mapping of ideas found in social science. The articles demonstrate common variables in their explanations, namely that power asymmetry and satisfaction with the status quo among regional partners are critical in determining why states integrate. The articles diverge in their explanations, with one emphasizing the similarities of institutionalized policies and the other the role a rising power, China, has in developing regional integration in the Western Hemisphere.

**Keywords:** asymmetric power; homogeneous institutions; power transition theory; regional integration; rising economic powers

## 1. Introduction

The international system has been witnessing an interesting phenomenon since the end of World War II. More and more states are joining organizations that seek to develop deep economic and sometimes political collaboration among their members. These regional collaborations are even outpacing global efforts. For example, the World Trade Organization's latest round of negotiations, the Doha Round, began in 2001 and has yet to make any significant movement towards successful completion.

Regional integration organizations (RIOs) are more than simple military alliances that the international system has had for thousands of years. What makes these sets of collaborations especially interesting is the multilateral sharing of state sovereignty within various policy areas. For example, some RIOs have an intergovernmental arrangement. They come together to make common decisions in certain policy areas. The enforcement of these decisions has traditionally depended on the member states. Some opt for creating regional institutions, but authority ultimately rests with the member states. Yet others adopt supranational institutional arrangements. Such regional institutions may make decisions alongside intergovernmental institutions or may even supersede the member states' national sovereign authority.

Why would states start moving away from the Westphalian state model, and why do we see variation in the level of regional integration worldwide? The literature on explaining regional integration goes back to the start of this phenomenon, namely the early days of what we now call the European Union (EU). Mitrany (1966) prescribed European integration as a method to prevent future continental wars. He argued that since wars were essentially a battle over resources, the development and regulation of major resources should move out of the hands of states and into the hands of functional organizations. Functionalism maintained that technological experts would guide resource use for the benefit of Europe as a whole and not for the needs of a single state. The lack of resource competition would lower the likelihood of war. Other scholars later transformed Functionalism from a prescriptive theory to a descriptive one in the development of Neofunctionalism (Haas

1958; Lindberg and Scheingold 1971). The theory also hypothesized integration processing through a functionalist logic; however, the result would be the formation of supranational institutions that would make decisions instead of the national governments. Instead of a web of functional organizations as prescribed by Mitrany, the Neofunctionalists described a process where a European state would form along a federalist framework.

Given the large expansion of RIOs and the wide variation in the depth of integration, we are seeing more generalized theories with systematic inferential testing of hypotheses that seek to explain and predict the development of regional integration (Soderbaum and Shaw 2003). The new approaches move away from a single RIO examination of integration, namely toward attempts to compare the development of European integration with others around the world. The idea is to see if common patterns emerge. This, of course, is possible because the number of cases has increased, giving us more data to analyze both across cases and across time. Such cases include newly started integration projects and projects that have increased the level of integration among member states. Additionally, under analysis are cases where negotiations failed, stagnated, or projects ended.

Many general theories start with similar assumptions, while others develop contradictory explanations. The volume of research has also hit a threshold where findings have become inconsistent with each other. However, we do see compilations of research that confirm earlier, similar hypotheses. Given the dramatic increase in the number of RIOs and their growing influence on political and economic affairs (Katzenstein 2005; Haftel and Thompson 2006; Gray and Slapin 2012), scholars should examine theory and empirical evidence more closely. One way would be to use a systematic methodology that deconstructs articles (and perhaps books) so we can more easily compare how work overlaps and where it diverges. Such a method is the systemist diagrammatic approach.

The two articles examined here using the systemist graphic approach provide general and specific explanations for regional integration. Systemist notation is followed in each of the forthcoming figures, and a full explanation of it appears in the introduction to this issue from Gansen and James (2023). The text in each figure is typed in upper- or lower-case characters. Upper-case characters are used for macro-level variables, while lower-case characters are used for micro-level variables. Each diagram also comes in double frames—the outer one refers to the environment, the inner one to the system.

In one of the two above-noted articles, Genna (2022) looks at the complex pathway of developing regional integration by using three variables: (1) the level of power asymmetry among member states; (2) how satisfied states are with the status quo among them; and (3) the homogeneity of their domestic institutionalized policies. The "status quo satisfaction" variable measures the correlation of United Nations voting patterns. Additionally, the degree of institutional homogeneity measures the similarity of policies in specific economic categories. The second article, also by Genna (2010), looks at the successes, failures, and stalemates of free trade agreement negotiations in the Western Hemisphere during the start of the 21st Century. This article explains negotiation outcomes by examining the level of power asymmetry and status quo satisfaction, but also introduces China's influence as a disrupting actor. Genna (2022) comes first given that it poses a generalizable theory and tests hypotheses quantitatively using a large number of cases. Genna (2010), while theoretically general, tests the hypotheses in a qualitative manner and therefore gives us a more detailed accounting of free trade negotiations in the Western Hemisphere.

## 2. A General Theory of Regional Integration

The first step is to explain, in general, why states in a particular geographic region integrate with each other economically. The answer also needs a dynamic mechanism to explain why states would stop at a given level of integration, such as a preferential trade agreement, or continue the process of increasing the fluidity of market transactions and their governance through collective political decision-making. Genna (2022) synthesized two approaches to determine a possible answer.

The first approach was the theoretical adoption of Power Transition Theory (Organski 1958; Organski and Kugler 1980) to the study of regional integration (Efird and Genna 2002). Power Transition Theory has a decades-long reputation for explaining interstate war and conflict, and it became a major alternative to neorealist (i.e., balance of power) theory and other frameworks of analysis. The further theoretical addition by Lemke (2002) to the regional level added to the possibilities of using Power Transition Theory to explain regional integration. Lemke theorized that power hierarchies among states are regional as well as global. The exploration of regional hierarchies opened the possibility of explaining the effects of power transitions at this level. In other words, the mechanism of global power transitions used to explain the onset of global war could also explain war at the regional level. Efird and Genna (2002) extended the idea further by examining if the mechanisms associated with the start of major powers or regional wars could explain why deep cooperation, similar to regional integration, might be achieved. The idea of establishing a satisfactory international status quo by a preponderant power and its closest allies has a direct overlap with what has been happening in regional economic relations since the end of World War II.

The explanation offered by Efird and Genna (2002) lacked a step that was answered in Feng and Genna (2003). What was missing was an understanding of the ease or difficulty of negotiations. In other words, do all negotiations have a common starting point? The answer would be 'no' if we assumed that the current state of institutionalized policies would vary across states within a region. They theorized and empirically supported the proposition that when institutionalized policies are similar across states, it is easier to negotiate regional economic agreements. Additionally, the process feedbacks to institutionalized policies, making them more homogenized over time to improve the economic gains from integration. Assuming that states wish to increase economic gains started by the early stages of integration, they would continue to negotiate further collaboration. This would entail homogenizing policies further so that non-tariff barriers do not impede collaboration already created or hinder future collaboration.

The synthesis of the two articles gives us a full potential pathway to explain regional integration. Figure 1 is a systemist diagram of Genna (2022). The system in the diagram is the discipline of International Relations. The macro and micro levels of the system correspond, respectively, to the discipline as a whole and individual scholars within it. The environment is the World Beyond.

Figure 1 begins in the World Beyond with the operational definition: 'REGIONAL INTEGRATION: ESTABLISHMENT OF REGULAR COLLECTIVE DECISION-MAKING AMONG STATES WITH THE INTENT OF DEVELOPING AND REGULATING MARKET FLOWS' (Haas 1958; Lindberg and Scheingold 1971). The article then asks the central question, 'WHAT EXPLAINS VARIATION IN THE LEVELS OF REGIONAL INTEGRATION WORLDWIDE?'

The article's focus shifts to the role that pre-negotiation policies play in predicting the success of negotiations. A review of the existing scholarship in International Relations points to 'RESEARCH GIVES IMPRECISE CAUSAL MECHANISMS, LEAVING UNANSWERED QUESTIONS REGARDING THE DEGREE TO WHICH HOMOGENEITY MATTERS AND THE IMPORTANCE OF DOMESTIC INSTITUTIONS.' The review also reveals that the 'LITERATURE DEMONSTRATES A COMPLEX PATH IN THE DEVELOPMENT OF REGIONAL INTEGRATION.' In other words, when looking at all the possible variables, there is a need to put the pieces together in a coherent model. When putting together the pieces, it becomes apparent that 'POWER TRANSITION THEORY PROVIDES AN ANSWER TO THE PUZZLE OF REGIONAL INTEGRATION'.

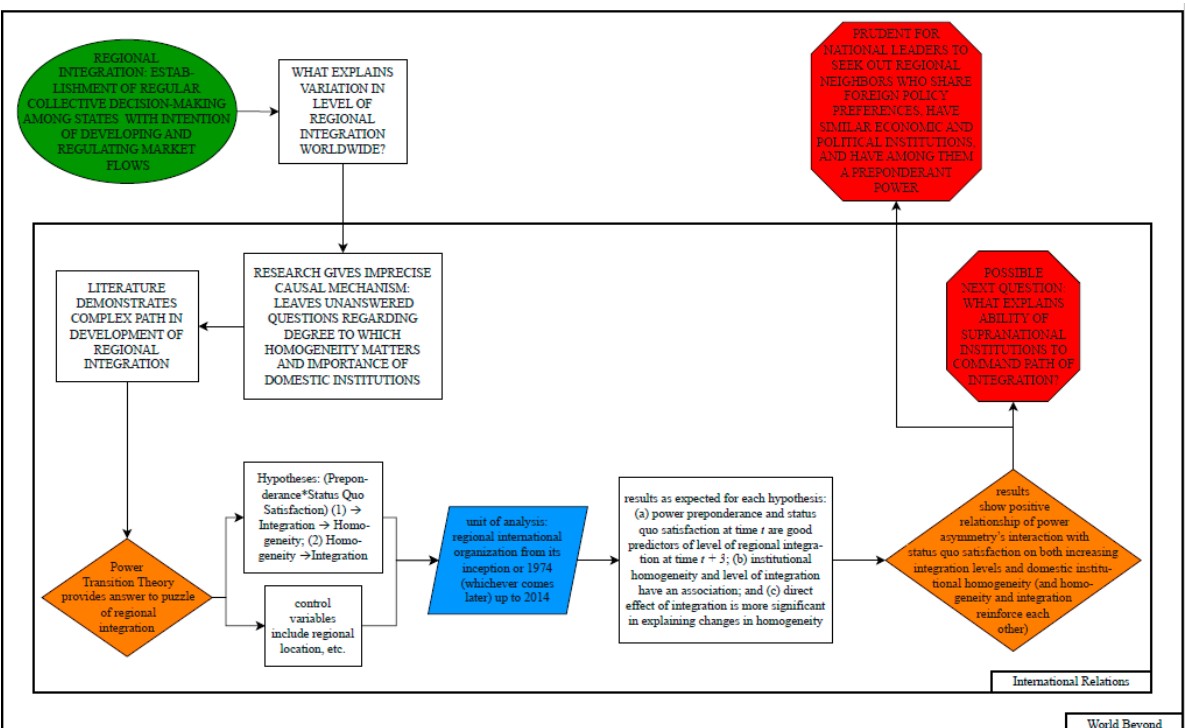

**Figure 1.** A pathway for regional integration (Genna 2022). Diagrammed by Sarah Gansen and Patrick James.

However, what are the theory's main variables, power asymmetry and status quo satisfaction, driving? The answer comes from Feng and Genna (2003). Many states, if not all, have institutionalized policies. Such policies are difficult to adjust since domestic actors have a strong stake in keeping them unchanged. A significant shift away from institutionalized policies may reduce the political survival of a state's government leadership. Therefore, the likelihood of successful negotiations increases when potential partners bring home to their domestic audiences agreements that require little change in domestic policies. In other words, the smaller the need for policy change, the more likely the domestic stakeholders will agree. This would be optimal among regional partners that have similar policies. The role of the preponderant power is to guide the negotiations by leveraging its economic size. As negotiations progress, the preponderant power can smooth out disagreements by offering incentives when domestic policy changes among partners are high.

In sum, the synthesis of the two theories—Power Transition and Institutional Homogenization—tells us that the presence of a preponderant regional power, status quo satisfaction, and the homogeneity of institutionalized policy preferences lead to the development of regional integration agreements. The theory also predicts a feedback loop in which policies undergo further homogenization in continued negotiations among the regional organization's member states. The systemist diagram notes the following: 'Hypotheses: (Preponderance*Status Quo Satisfaction) → (1) Level of Integration → Homogeneity; (2) Homogeneity → Level of Integration' and 'control variables include regional location, etc.' The notation "Preponderance*Status Quo Satisfaction" signifies an interaction term. The data analysis uses the 'unit of analysis: regional international organization from its inception or 1974 (whichever comes later) up to 2014'.

The data support the multiple hypotheses: 'results as expected for each hypothesis: (a) power preponderance and status quo satisfaction at time $t$ are good predictors of level of regional integration at time $t + 3$; (b) institutional homogeneity and level of integration have an association; and (c) the direct effect of integration is more significant in explaining changes in homogeneity'. In other words, power preponderance and status quo satisfaction do predict the level of regional integration three years out. Homogeneity in institutionalized



policies also predicts the development of regional integration. There is, furthermore, a direct effect of greater regional integration on promoting homogeneity. This finding leads to the following conclusion: 'results show a positive relationship between power asymmetry's interaction with status quo satisfaction on both increasing integration levels and domestic institutional homogeneity (and homogeneity and integration reinforce each other).'

This brings us to two important implications. One is for International Relations, and the other is for the World Beyond. First, with regard to International Relations, a significant question arises: is it possible for the level of integration among states to produce institutions that begin to take over the decision-making process? In other words, 'POSSIBLE NEXT QUESTION: WHAT EXPLAINS THE ABILITY OF SUPRANATIONAL INSTITUTIONS TO COMMAND THE PATH OF INTEGRATION?'.

Second, in the World Beyond, it is 'PRUDENT FOR NATIONAL LEADERS TO SEEK OUT REGIONAL NEIGHBORS WHO SHARE FOREIGN POLICY PREFERENCES, HAVE SIMILAR ECONOMIC AND POLITICAL INSTITUTIONS, AND HAVE AMONG THEM A PREPONDERANT POWER'. The recommendation states that domestic leaders need to find optimal conditions for political negotiations that may not follow economic theories, such as comparative advantage. Should homogeneity, at least to a small degree, not be present, it will require the preponderant power to do more heavy lifting to get status quo allies to join regional integration organizations. Therefore, a region would need a large power asymmetry to bring heterogeneous states together. Another factor would be a large degree of status quo satisfaction. The optimal condition would be to have both. This way, regional integration can begin when homogeneity is low in order for the process of greater homogenization to develop. Of course, it may be that some geographically connected countries do not have these optimal conditions, which would also add to the explanation revealed by the theory regarding the uneven integration of regions around the world.

## 3. Western Hemisphere Free Trade Agreements at the Start of the 21st Century

The second systemist diagram outlines the argument and evidence in Genna (2010). The article explains why some Western Hemisphere trade negotiations were successful while others resulted in stalemates at the start of the 21st century. Following insights from Power Transition Theory, open trade arrangements develop when larger regional economies offer incentives to smaller economies. Smaller economies are attracted to the arrangements so they can access larger economies. While the theory predicts and the evidence supports the strong likelihood of this occurring under extreme asymmetry, it is not clear what actions would be taken by middle-sized economic powers. The article conducts a qualitative analysis to seek an answer to this question.

In Figure 2, which depicts the argument from Genna (2010), the Western Hemisphere is the system. The micro and macro levels of that system correspond, respectively, to actors and interactions among them. The International System is the environment. A pathway begins in the International System with 'PROLIFERATION OF FREE TRADE AGREEMENTS (FTA): INCREASINGLY A NORM IN INTERNATIONAL POLITICAL ECONOMY (IPE)'. However, not all negotiations are successful. Therefore, we need to 'SEEK TO EXPLAIN WHY CERTAIN FTA NEGOTIATIONS STALEMATE'.

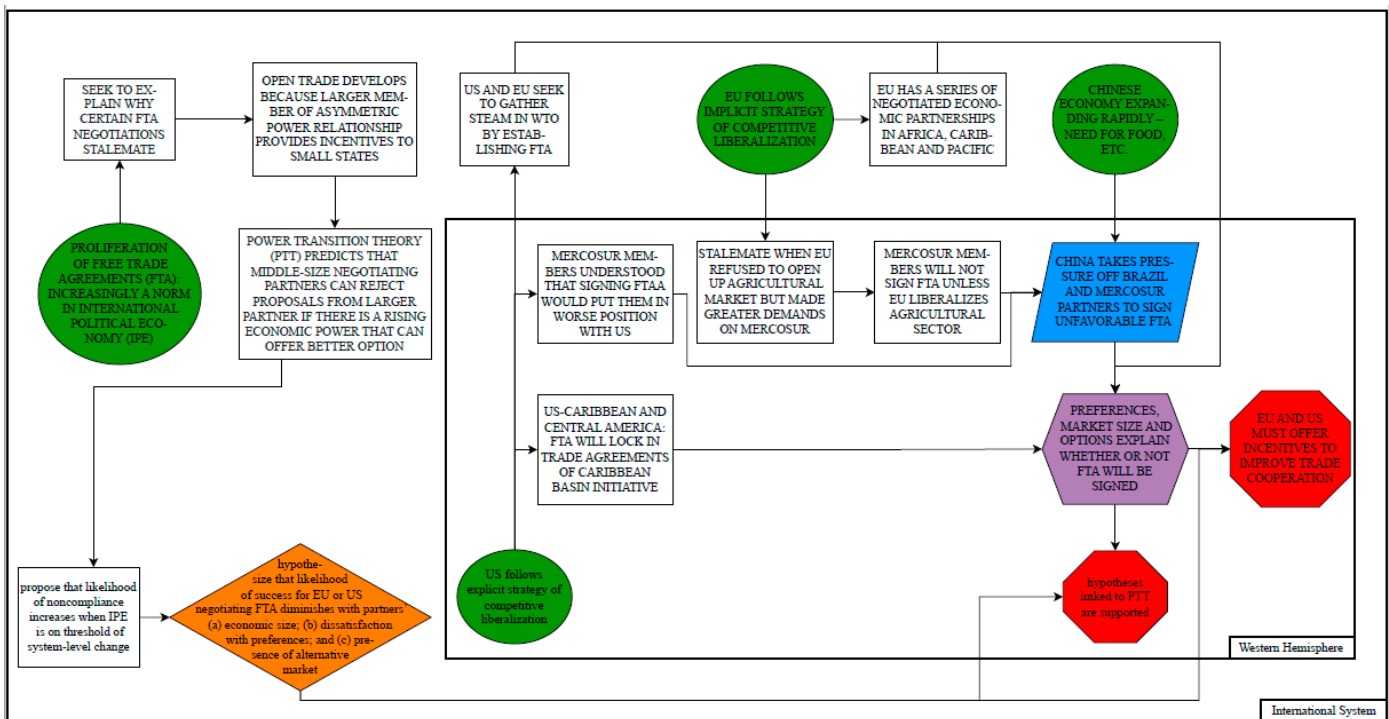

**Figure 2.** Economic size and the changing international political economy of trade: The development of Western Hemisphere FTAs (Genna 2010). Diagrammed by Sarah Gansen and Patrick James.

The next step is to examine three variables: asymmetry, preference satisfaction, and the presence of an alternative open trade arrangement. As mentioned, asymmetry has benefits for both large and small economies. Small economies obtain access to larger markets. States with large economies gain a growing number of currently satisfied status quo states within the organization and, due to economic interdependence, are more likely to accept its future preferences should things change. To apply the Power Transition terminology, it will use economic ties to establish a status quo that would be acceptable to smaller states. The last variable is the presence of an alternative trade arrangement with another large power. This other large power would be the rising power that could challenge the current preponderant power. The alternative arrangements would be preferable to the smaller states if the conditions set by the rising power had greater advantages for the smaller states.

This is where the size of smaller states matters most. Very small states may not have much choice because their size makes them vulnerable. Being small, their production options are limited, meaning that they need imports to sustain their economies. In addition, as mentioned, their small domestic market size will quickly saturate, requiring firms to seek out markets, preferably much larger ones, in order to achieve economies of scale. Therefore, if a state with a large market is willing to negotiate entry, then it has the capability to include conditions. Such conditions will allow the larger state to produce a status quo in its favor. The thoughts summarized in the diagram appear as 'OPEN TRADE DEVELOPS BECAUSE A LARGER MEMBER OF THE ASYMMETRIC POWER RELATIONSHIP PROVIDES INCENTIVES TO SMALL STATES'.

When negotiations involve middle-sized powers, then things become more complicated. 'POWER TRANSITION THEORY (PTT) PREDICTS THAT MIDDLE-SIZE NEGOTIATION PARTNERS CAN REJECT PROPOSALS FROM LARGER PARTNERS IF THERE IS A RISING ECONOMIC POWER THAT CAN OFFER A BETTER OPTION'. Rising powers can have the same preferences as the current SQ power or different ones. Power competition (and possibly war) increases when preferences are different. According to Power Transition Theory, the US took over as the SQ power from the UK post-WWII because they agreed on most things. Today, middle-sized powers can take advantage of the competition

created by the rise of China. Being middle-sized means a state is not as dependent on trade as relatively small economies. They possess more production variability and a larger domestic market for sales. If a preponderant power is the only option, then it, similar to the smaller powers, will need to acquiesce to conditions if it wants access to the larger market. The presence of a rising power changes the calculations, especially if the rising power requests little to no conditions on trading arrangements. The middle-sized powers may therefore have less pressure to sign an agreement with the preponderant power. In sum, we 'propose that the likelihood of noncompliance increases when IPE is on the threshold of system-level change'. The systemist diagram highlights the article's theoretical section and 'hypothesizes that the likelihood of success for EU or US negotiating FTAs diminishes with partners' (a) economic size, (b) dissatisfaction with preferences, and (c) the presence of alternative markets'.

The article tests the hypotheses using four case studies. The four cases offer a "natural experiment" where we can see within the same timeframe evidence of the theory's mechanisms at work. We have two large economic powers seeking to establish trade regimes in the Western Hemisphere: the EU and the US. Both economic powers are attempting negotiations with small economic powers in the Caribbean basin (islands plus Central America) as well as middle-sized powers in South America. All the negotiations are occurring while trade with China is on the rise. The conditions are ideal for providing evidence that small states will tend to acquiesce to larger ones, while middle-sized ones have greater agency when a rising power is present.

The first two cases involve the European Union (EU) and are shown in Figure 2 as a systemist diagram starting at the macro level of the International System: 'EU FOLLOWS IMPLICIT STRATEGY OF COMPETITIVE LIBERALIZATION'. The EU's implicit strategy of competitive liberalization seeks to develop an international trading system according to its preferences (i.e., within the World Trade Organization (WTO)). Thus, the 'EU HAS A SERIES OF NEGOTIATED ECONOMIC PARTNERSHIPS IN AFRICA, THE CARIBBEAN, AND THE PACIFIC'. The economic partnerships are with African, Caribbean, and Pacific (ACP) states, all of which were former colonies of some EU member states. The agreements include multiple conditions that are not favorable to ACP heavy manufacturing but are favorable to some sectors of ACP agriculture. In addition, the EU requires significant noneconomic conditions, such as adhering to the rule of law and especially human rights. Given the large asymmetries between the EU and the ACP states, the negotiations were successfully completed.

The second EU case is an attempt to sign a free trade agreement with the MERCOSUR[1] countries. It serves as an illustration of a stalemate negotiation: 'STALEMATE WHEN THE EU REFUSED TO OPEN UP THE AGRICULTURAL MARKET BUT MADE GREATER DEMANDS ON MERCOSUR'. These negotiations stagnated because the EU did not want to open its agricultural sector to the MERCOSUR countries but instead wanted MERCOSUR to open up its high-tech sectors. The next step moves toward rejection of the EU proposal: 'MERCOSUR MEMBERS WILL NOT SIGN AN FTA UNLESS THE EU LIBERALIZES THE AGRICULTURAL SECTOR'. The MERCOSUR member states would not sign under these terms because agricultural products were their main exports. The two major members, Argentina and Brazil, both middle-sized economies, saw that the arrangement would be especially harmful for their interests.

This is where the rising power, China, comes in: 'CHINESE ECONOMY EXPANDING RAPIDLY—NEED FOR FOOD, ETC'. By desiring to purchase MERCOSUR's agricultural products, 'CHINA TAKES PRESSURE OFF BRAZIL AND MERCOSUR PARTNERS TO SIGN AN UNFAVORABLE FTA'. The alternative growing Chinese market would take the MERCOSUR exports and therefore allow time for a more favorable agreement with the EU.

The remaining two cases involve negotiations between the US and Latin America. The 'US follows an explicit strategy of competitive liberalization' to configure the global trade regime according to its preferences. Therefore, together, the 'US AND EU SEEK TO GATHER STEAM IN THE WTO BY ESTABLISHING FTAS'. Similar to the EU cases,

the negotiations with the smaller Latin American states were successful, while the negotiations that included the middle-sized states were not. Successful negotiations culminated in The Dominican Republic-Central America-United States Free Trade Agreement (CAFTA-DR): 'US-CARIBBEAN AND CENTRAL AMERICA: FTA WILL LOCK IN TRADE AGREEMENTS OF THE CARIBBEAN BASIN INITIATIVE'. The small states approved the agreement because CAFTA-DR maintained the benefits associated with the Caribbean Basin Initiative.

Figure 2 then turns to the second case, which focuses on the negotiations that would have developed the Free Trade Area of the Americas had the parties ended negotiations unsuccessfully: 'MERCOSUR MEMBERS UNDERSTOOD THAT SIGNING AN FTA WOULD PUT THEM IN A WORSE POSITION WITH THE US'. Similar to the EU-MERCOSUR FTA, the US would not open markets to allow competition from Brazilian and Argentinian agriculture but wanted access to MERCOSUR's capital-intensive product market. As in the negotiations with the EU, 'CHINA TAKES PRESSURE OFF BRAZIL AND MERCOSUR PARTNERS TO SIGN AN UNFAVORABLE FTA'. The middle-sized states had access to Chinese markets, and they were therefore under little pressure to conclude negotiations with the prevailing US preferences. The result was the suspension of trade talks.

The systemist diagram ends with the following points: First, the qualitative analysis demonstrated that 'hypotheses linked to PTT are supported'. Next, 'PREFERENCES, MARKET SIZE, AND OPTIONS EXPLAIN WHETHER OR NOT AN FTA WILL BE SIGNED'. The alternative options offered by the rising power (China) can explain the failures of the Western Hemisphere free trade negotiations with the middle-sized powers. Last, the 'EU AND US MUST OFFER INCENTIVES TO IMPROVE TRADE COOPERATION'. Negotiating with middle-sized powers requires an understanding of the agency that the rise of China provides.

## 4. Systematic Synthesis

Among the systemist methods, systematic synthesis is appropriate for engaging Figures 1 and 2 with each other. Systematic synthesis refers to the comparison of diagrams that represent studies in relatively close proximity to each other in terms of subject matter and method (Gansen and James 2023). The goal of such comparative analysis is to identify points of (dis)agreement—a synthesis of what can be gleaned from the two diagrammatic expositions in combination with each other.

A systematic synthesis of the two articles (Genna 2010, 2022) brings forth interesting conclusions. First, both articles have a common starting point: states join and further develop RIOs for their own self-interests and design them in such a way as to increase their institutional value (Koremenos et al. 2001). Second, the differentiated roles of states are important: states play different roles depending on their economic size. Regional preponderant powers have the state capacity to guide decisions, provide incentives, and promote rules. Small states often acquiesce to large states given these incentives. Middle-sized states have greater agency when alternative options are available. Third, the preferred conditions found in regions—the status quo—matter. Collaborating with a preponderant power can be beneficial or costly, depending on the status quo the large power wishes to create. If the partnership becomes costly due to the status quo, then small and medium-sized states may opt not to form it or wish to develop further regional integration. If the partnering states view the status quo as beneficial, then they are more likely to form and develop integration.

Systemist diagrammatic analysis also exposed important differences between the two articles. Genna (2022) argued and empirically supported the idea that the driving mechanism behind integration is the homogeneity of institutionalized policies. The more similar the policies among states, the more likely they are to integrate because less change is expected. Power asymmetry and status quo satisfaction drive the homogeneity and integration processes. Genna (2010) introduced the disruptive role of the rising power. One of the original goals of Power Transition Theory was to show how changes in the

international system occur when a state begins to ascend and attempts to reform the international status quo. With the rise of China, we are witnessing China's disruptive influence on both US and EU efforts to command the global trade regime. China's economic expansion in the early 2000s provided an alternative market for MERCOSUR agricultural exports. So much so that it had the capability not to sign on to the Free Trade Area of the Americas. Additionally, it was able to wait out the EU during FTA negotiations. As a postscript, the EU did eventually accept MERCOSUR agricultural products in the FTA, which the parties signed after twenty years of negotiations. However, the parties have yet to ratify the agreement.

## 5. Conclusions

The systemist diagrammatic method allows scholars to map out the ideas and findings found in social science. By doing so, faculty and students alike can understand what this research paper is attempting to explain in a "no frills" manner. The deconstruction and mapping exercise is also a helpful tool in building literature reviews because one can quickly see the similarities and differences between research agendas. One can then tease out the gaps and contradictions in order to move the body of knowledge forward.

The examination of Genna 2010 and 2022 demonstrates the systemist method's ability to piece together the arguments associated with the development of regional integration institutions. Central to both articles is the need for regional asymmetric power distribution and satisfaction with the status quo that the preponderant power has developed and will continue to maintain. Both are necessary conditions but not entirely sufficient on their own. Genna (2010) demonstrates that the existence of a rising power will begin to disrupt the preponderant power's attempts to conclude successful free trade negotiations with middle-sized powers. China's need for raw resources gave the middle-sized powers in MERCOSUR the agency to reject offers or stalemate negotiations. Genna (2022) added another condition, namely the degree to which institutionalized domestic policies are similar among the negotiating parties. The greater the homogeneity of policies among states, the greater the likelihood that negotiations can conclude successfully because states have little domestic change to sell back home. In addition, similar starting points could advance negotiations in the future since change will not likely be large and deviate from existing starting norms.

**Funding:** This research received no external funding.

**Institutional Review Board Statement:** No IRB review required.

**Informed Consent Statement:** Not applicable.

**Data Availability Statement:** Data available upon request.

**Conflicts of Interest:** The author declares no conflict of interest.

## Notes

[1]     MERCOSUR is the Spanish acronym for the Common Market of the South made up of Argentina, Brazil, Paraguay, and Uruguay.

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
