# Peer review of "When States Regionally Integrate: How Relative Economic Size and Institutional Homogeneity Matter"

_socsci, doi:10.3390/socsci12070397_

Round 1

Reviewer 1 Report

This article on ‘When States Regionally Integrate’ provides a nice synthesis of some literatures linking the PTT and FTA issues. Some parts raised further questions. I provide them below, which might help the author in a re-write.

1.      53-55: I don’t understand why real world positive events give more data. Don’t non-events yield data too?

a.      165-68: why are we looking only at RIOs? Are we selecting on the DV?

b.     Not clear to me the difference of RIO (unit of analysis) and the DV ‘regional integration’?

2.      Unclear how we get from homogenous policies to SQ. How is SQ measured?

3.      Did the test do an interaction of preponderance and SQ? I understood later in the text that this is a yes: consider making this clearer earlier.

4.      225: What future preferences? Why the future and not now? Why trust the future?

5.      Why middle-sized powers different? Why can’t small states also deal with rising hegemon?

6.      249: why does the rising power request fewer conditions than the SQ power? What are these conditions? Why can’t the SQ power also have no conditions?

7.      292: what is “pressure”? Aren’t countries free to say ‘no’?

8.      293: not clear how time comes in. Do we need “and therefore allow time for a more  favorable agreement with the EU” It seems like something is being implied, and I don’t know what is.

9.      Should explain the “Caribbean Basin initiative” on first mention. Doesn’t this give the countries unilateral tariff free access to the US market? I think there are no conditions here.

10.  Why not alternative explanation that smaller states simply do not threaten European farmers because they cannot produce as much agricultural competition for European farmers?

11.  309: what is ‘worse position with US’? Is relative gains competition being implied here? If so, why?

12.  312: what pressure?

13.  314: domestic pressure? 316+ confusing because ‘PREFRENCES, MARKET SIZE’ does not follow ‘hypotheses linked’ as diagramed.

14.  ‘EU AND US . . .” Why must EU and US offer incentives? It seems as if some goal besides economic self-interest is being implied here.

15.  Why can’t small states make agreements with China?

16.  Can states make agreements with US, EU and China at the same time?

17.  What about everything explained by China wants agricultural products, and Europe and EU do not? Therefore, medium states with farmland have something to offer China.

a.      Says this at the end 376, but phrases it as fitting the PTT? Isn’t this more an alternative explanation, perhaps a threat to the natural experiment?

18.  358: unclear what ‘wait out’ means. It seems like something is being implied.

19.  Typos/grammar

a.      97, 283

b.     144: “PROVIDES ANSWER” should it be “PROVIDES AN ANSWER”?

c.      Need grammar checks for ‘the’

d.     198: ‘revealed’?

fine

Reviewer 2 Report

The paper is an interesting attempt to summarize important findings in the discussion of Genna (2022) and Genna (2010). The chosen two articles by Genna complement each other by building out theoretical and empirical aspects of the arguments around the formation of regional integration organizations (RIO). The systemist diagrammatic approach is simple in its conception but appropriate for the use of providing graphical depictions of International Relations. The paper also gives a comprehensive analysis with a potential policy vision at the outset, with clear policy-based narrative in its core, and  advisory at the end.

Some minor comments on how the paper could provide a clearer presentation of the method applied are:

  1.  The study should explain further the systemist notation used to develop the analysis:  how many types of variables are used, the types of connections between and among macro and micro variables,  how initial and terminal variables play out in the International System and World Beyond fin Figure 1 as well as Western Hemisphere and International System in Figure 2, why some variables converge or follow their separate pathways. 

          2. The diagram in Figure 1 might need to explore greater use of the results related to the operation of different variables, at different levels of integration, homogeneity, and status quo satisfaction. This can be combined with a reasoned set of policy implications for RIO. This is important so as to better visualize the impact of each variable. 

      3. Figure 2 needs to include other control variables that explain the successes and stalemates of trade negotiation. Other salient variables could be the the emergence leaders of middle sized states'  and their opposition to FTAs with the US, how labor and environmental standards in China are different from those in the US and the EU,  recent slowdown in EU integration, the rising power of other countries, etc.

4. This paper uses the power transition theory to explain well the relationship between asymmetric power, evaluations of the status quo, regional hierarchies to account for cooperation.  An extended study or argument with the systemist diagrammatic approach could be made to explain conflicts and economic divergence.
